# Prevalence and Persistence of Ceftiofur-Resistant *Escherichia coli* in A Chicken Layer Breeding Program

**DOI:** 10.3390/ani13010090

**Published:** 2022-12-26

**Authors:** Meina Liao, Jiaen Wu, Yafei Li, Xiaoqing Lu, Huizhen Tan, Shanshan Chen, Wenqing Huang, Xinlei Lian, Jian Sun, Xiaoping Liao, Yahong Liu, Saixiang Feng, Rongmin Zhang

**Affiliations:** 1Guangdong Provincial Key Laboratory of Veterinary Pharmaceutics Development and Safety Evaluation, South China Agricultural University, Guangzhou 510642, China; 2Guangdong Laboratory for Lingnan Modern Agriculture, Guangzhou 510642, China; 3National Risk Assessment Laboratory for Antimicrobial Resistance of Animal Original Bacteria, South China Agricultural University, Guangzhou 510642, China; 4Institute of Quality Standard and Monitoring Technology for Agro-Products, Guangdong Academy of Agricultural Sciences, Guangzhou 510640, China; 5College of Veterinary Medicine, South China Agricultural University, Guangzhou 510642, China

**Keywords:** longitudinal monitoring, ceftiofur resistance, ST101 *E. coli*, breeder farm

## Abstract

**Simple Summary:**

This is a study associated with the prevalence and persistence of ceftiofur, the third-generation cephalosporin antibiotic-resistant *Escherichia coli* in a chicken layer breeding farm. Notably, epidemiological studies in poultry have been primarily focused on single breeding periods and may have underestimated the maintenance of the ceftiofur-resistant *Escherichia coli* in different breeding periods from layer hens. Our results showed that the detection rates of the ceftiofur-resistant *Escherichia coli* fluctuated across different breeding periods, and the ST101 ceftiofur-resistant *Escherichia coli* was the most prevalent and persistent sequence type across the breeding periods. This study contextualized ceftiofur resistance in different breeding periods, monitoring which can have important implications for food animals.

**Abstract:**

We determined the longitudinal persistence of ceftiofur-resistant *Escherichia coli* from a chicken breeding farm in China. A total of 150 samples were collected from 5 breeding periods in a flock of layer hens, and the prevalence of ceftiofur-resistant *E. coli* fluctuated across the 5 chicken breeding stages: eggs, 3.33%; growing period, 100%; early laying period, 36.7%; peak laying period, 66.7% and late laying period, 80%. The most prevalent ceftiofur resistance genes were *bla*_CTX-M-55_, *bla*_CMY_ and *bla*_NDM_, and ST101 was the most prevalent and persistent sequence type across the breeding periods. Our results indicated that this breeder flock was heavily contaminated by ST101 ceftiofur-resistant *E. coli* and that its presence should be intensively monitored on chicken farms.

## 1. Introduction

The increasing prevalence of extended spectrum β-lactamase (ESBL) Gram-negative bacteria in food-producing animals poses a great challenge for animal husbandry and especially the poultry industry [1,2]. Of particular concern is the third-generation cephalosporin, ceftiofur, that was developed strictly for veterinary use [3]. In China, ceftiofur was commonly used in all food animals, including poultry, to control and prevent bacterial infections [4], although its use is restricted to swine and cattle in the European Union [5]. Ceftiofur effectiveness has been compromised by ESBLs, ampicillin class C (AmpC) β-lactamases and carbapenemases [6]. The CTX-M type is currently the predominant cephalosporin-degrading enzyme in *Escherichia coli* of poultry origin and the CTX-M-55 subtype is the most common [7,8,9,10,11,12]. A previous study implicated CTX-M-55 as the major ESBL gene carried by *E. coli* (EC) chicken isolates in an epidemic study in four Chinese provinces from 2015–2019 [7]. In a Korean nationwide study, CTX-M-55 was also the most prevalent ESBL-EC across 21 poultry farms, 20 retail stores, 6 slaughterhouses and 111 workers [10]. Meanwhile, *bla*_CTX-M-55_ frequently co-occurs with other antibiotic resistance genes (ARGs) including *mcr*, *oqxAB*, *fosA3* and *floR* that respectively confer resistance to quinolones, fosfomycin and amphenicol [13,14,15]. This high prevalence of ESBL-EC poses a serious threat to the food animal production industry.

Epidemiological studies in poultry have been primarily focused on single breeding periods and may have underestimated disease impacts [11,16]. There are five chicken breeding growth cycles: the brooding (BP), growing (GP), early laying (EL), peak laying (PL) and late laying (LL) periods [17]. There have been few studies that traced persistence of ceftiofur-resistant *E. coli* across the chicken growth cycle, and these were carried out with broiler chickens. For example, the prevalence of ESBL/AmpC-producing *E. coli* displayed an increasing trend across three sampling times within one broiler flock fattening period [18]. A longitudinal monitoring study of ESBL/AmpC-producing and fosfomycin-resistant *E. coli* in 8 broiler farms indicated the presence of ESBL-EC between days 20 and 25 that then increased in the fattening period at days 36 to 38 [19]. In chicken breeding farms, only a single study has analyzed ARG diversity and abundance in different breeding periods [17]. In nine farms in Guangdong Province, the ranked prevalence of ARG abundance in manure samples was BP > LL > GP > EL> PL [17]. However, the prevalence and persistence of ceftiofur-resistant *E. coli* was not investigated.

In the current study, we tracked the prevalence and maintenance of ceftiofur-resistant *E. coli* isolated from different breeding stages of a chicken flock in a Chinese breeder farm and found that ST101 ceftiofur-resistant *E. coli* was the most prevalent subtype in the chicken farm and persisted throughout layer breeding.

## 2. Materials and Methods

### 2.1. Sample Collection

Samples (150) from chickens across five breeding periods [17] were collected for this study from a breeder farm in Guangdong province, China, as previously reported [17], In brief, 30 samples per stage were taken from eggs and cloacal swabs from GP, EL, PL and LL. Each cloacal sampling was obtained using a sterile swab that was inserted into the chicken cloaca for at least three seconds (Appendix A). Samples were stored at 4 °C prior to analysis and then inoculated within 24 h (see below). The sampling of animals was conducted in accordance with the principles of the South China Agriculture University Animal Ethics Committee (Number: 2019g004).

### 2.2. Questionnaire

Participants providing samples from chicken farms were invited to answer a questionnaire to indicate the types of antimicrobials that were in use at the time of sample collection.

### 2.3. Bacterial Isolation and Identification

Each cloacal swab sample was spread on MacConkey plates containing 8.0 mg L^−1^ ceftiofur as previously described [20,21]. Each egg was massaged with 8 mL of brain heart infusion (BHI) broth in an aseptic plate for one minute and then spread onto MacConkey plates containing 8.0 mg L^−1^ ceftiofur and incubated for 18 h at 37 °C. One colony per plate was selected for Gram staining and species identification using a 16S rDNA sequence analysis as previously described [22].

### 2.4. Antibiotics Susceptibility Testing

Antimicrobial susceptibility testing was performed using the broth microdilution method and minimal inhibitory concentrations (MIC) were interpreted according to guidelines of the Clinical and Laboratory Standards Institute veterinary antimicrobial susceptibility testing standard (https://clsi.org/standards/products/veterinary-medicine/documents/vet01s/ and https://clsi.org/standards/products/microbiology/documents/m02/ (accessed on 3 November 2021)). All *E. coli* isolates were tested for susceptibility to ceftazidime, aztreonam, meropenem, gentamicin, ciprofloxacin, colistin and ceftiofur. *E. coli* ATCC 25922 was used for quality control.

### 2.5. Molecular Typing of Ceftiofur-Resistant E. coli

Enterobacterial repetitive intergenic consensus-PCR (ERIC-PCR) was used for sequence typing as described previously [23]. ERIC-PCR patterns were defined using a cut-off of 80% identity between DNA band patterns [24].

### 2.6. Sequence Assembly and Annotation

One isolate from each unique ERIC-PCR pattern was further selected for whole genome sequencing (WGS). Genomic DNA was purified using a commercial kit (Tiangen, Beijing, China). DNA libraries were constructed using the NEXT Ultra DNA Library Prep kit (New England Biolabs, Beverly, MA, USA) and WGS was performed using the Illumina HiSeq 2500 system (Novogene, Guangzhou, China). The draft genome was de novo assembled using SPAdes version 3.9.0 [25]. Multilocus sequencing typing (MLST), plasmid incompatibility (Inc) groups and ARGs were identified using MLSTcheck (https://github.com/sanger-pathogens/mlst_check, accessed on 15 December 2022) and ABRicate (https://github.com/tseemann/abricate, accessed on 12 May 2020), respectively.

### 2.7. Phylogenetic Analysis of Ceftiofur-Resistant E. coli

The molecular phylogeny of the 24 ceftiofur-resistant *E. coli* we identified was traced using 1145 isolates from the GenBank database to construct a phylogenetic tree using Parsnp software of the Harvest suite [26]. In brief, a total of 774,435 assembled bacterial whole genome sequences were downloaded from the GenBank database as of 7 November 2020 [27]. The ARGs carried by *E. coli* isolates with isolation information were identified using standalone BLAST analyses against the ResFinder [28]. The *E. coli* isolates we used from the public database possessed at least one of the ceftiofur resistance genes (CTX-M-55, CMY-2 and NDM-5) that was also present in our 24 study isolates. This analysis resulted in a total of 1145 *E. coli* isolates that were used for further study (Appendix A). The variant call format (VCF) file for the variants identified with Parsnp was then used to determine pair-wise single nucleotide polymorphism (SNP) distances between core genomes. To estimate the *E. coli* population structure, BAPS v6.0 software [29,30] and the module hierBAPS [31] was applied to the data to fit lineages to genome data using nested clustering. BAPS groups were assigned based on SNPs in the core genome of the *E. coli* strains.

### 2.8. Data Availability

Genome assemblies of the 24 sequencing strains in this study were deposited in GenBank under BioProject accession number PRJNA857571.

## 3. Results and Discussion

### 3.1. The GP Breeding Stage Is Associated with High Level Ceftiofur Resistance

We obtained 86 ceftiofur-resistant *E. coli* isolates (57.3%) from our 150 samples and their prevalence varied across the 5 chicken breeding periods. Only a single ceftiofur-resistant *E. coli* was recovered from egg samples. All GP samples (100%) were positive for ceftiofur-resistant *E. coli* and the prevalence decreased at EL (36.7%) but then increased in PL (66.7%) and LL (80%) (Appendix A). The high GP prevalence was consistent with a prior study of ESBL/AmpC-producing *E. coli* that remained at 100% prevalence in broiler chickens from weeks 3–5 [32]. Our results indicated that the high GP prevalence of the ceftiofur-resistant *E. coli* was most likely associated with the administration of β-lactam antibiotics during this period [33] and this was consistent with the antimicrobials (amoxicillin and ceftiofur) used on the farm (Appendix A). All these 86 ceftiofur-resistant *E. coli* isolates were resistant to ceftiofur and ceftazidime, 78 were also ciprofloxacin-resistant and 85 and 60% were resistant to aztreonam and gentamicin, respectively (Appendix A). However, only 11 isolates exhibited resistance or intermediate resistance to meropenem, and most of the isolates remained susceptible to colistin (94.2%). Moreover, higher MICs for aztreonam, gentamicin, ciprofloxacin and ceftiofur were found in the isolates from the GP period than from other breeding stages. These isolates were extremely highly resistant (MIC ≥ 256 mg L^−1^) to ceftiofur, gentamicin and ciprofloxacin and indicated that the GP period generated the most serious resistance phenotypes.

### 3.2. Molecular Characterization of Ceftiofur Resistant E. coli

ERIC-PCR was successfully performed on 84 ceftiofur-resistant *E. coli* (2 failed) resulting in 22 patterns (A to V) (Appendix A). Clonal spread was suggested in isolates that shared a >80% identify for each unique ERIC-PCR pattern. Each pattern was shared by a range of 1–20 isolates and the most common were types R (23.8%; 20/84), E (8.3%; 7/84) and J (8.3%, 7/84). One isolate from each of the 22 unique patterns and the 2 ERIC-PCR failed isolates (24 total) were selected for further WGS (Appendix A and Appendix A). We identified nine known STs including ST101, ST602, ST746, ST7611, ST48, ST206, ST1158, ST1485 and ST354. ST101 predominated, and this was consistent with the frequent identification of ST101 in *E. coli* isolates from farms, slaughterhouses and food markets [34]. In addition, we identified <35 SNPs in each of the 22 unique ERIC-PCR patterns, suggesting a clonal pattern of spread [35,36,37]. We further identified 9 distinct clones based on the SNP analysis, but 22 clonal relationships using ERIC-PCR, indicating SNP-based analysis could more precisely distinguish the clonal relationships (Figure 1).

In our 24 samples selected for WGS analysis, we found 12 ST101 *E. coli* isolates harboring ESBLs and these were present in all 4 stages of layer breeding. The other STs were detected discontinuously or continuously in only 2–3 stages. For example, ST354 persisted across three consecutive periods, ST206 for three discontinuous periods while ST602, ST7611, ST48, ST1158 and ST1485 were only observed in two stages each (Figure 1). These data suggested that ST101 *E. coli* was the predominant and persistent clone during the layer breeding and agreed with our previous study using randomly collected samples from a Chinese poultry production chain [22]. The long-term persistence of ST101 *E. coli* during layer breeding was likely to be the reason for the high prevalence of ST101 *E. coli* on the chicken farm. This phenomenon may be associated with an advanced fitness of ST101 *E. coli* carrying ceftiofur resistance genes in chickens and should be further explored.

### 3.3. Phylogenetic Analysis with GenBank Ceftiofur-Resistant E. coli

Interestingly, WGS analysis of our isolates indicated only three ARGs that were responsible for the ceftiofur resistance we found: the ESBL *bla*_CTX-M-55_, the AmpC *bla*_CMY-2_ and the carbapenemase *bla*_NDM-5_). We compared the WGS data to *E. coli* strains in GenBank (as of 7 November 2021) that carried at least 1 of these genes and identified 1145 *E. coli* strains (Appendix A). The core genomes of isolates sharing <35 SNPs were grouped and identified as clonally spread [35,36,37] and one isolate from each group was randomly selected for further study. This resulted in 580 isolates that were selected for phylogenetic tree construction and consisted of 571 GenBank samples and 9 samples from the current study (Appendix A).

The 580 *E. coli* isolates were primarily of human origin and nearly half originated in China. A large ST diversity was also found that was dominated by ST167 (5.5%), ST10 (5.0%), ST48 (4.5%), ST156 (3.8%), ST617 (2.9%), ST410 (2.8%) and ST101 (2.6%). All isolates in the phylogenetic tree were classified into 7 clades sharing a total of 96,249 SNPs. The most prevalent ceftiofur resistance genes were *bla*_CTX-M-55_ (57.2%, 332/580), *bla*_NDM-5_ (32.9%, 191/580) and *bla*_CMY-2_ (5.5%, 32/580) (Figure 2). This pattern was similar to that in the current study, where CTX-M-55 predominated in the 24 *E. coli* (13/24) we sequenced (Figure 1).

Our study was limiting in that (1) the epidemiology of the long-term persistence of ceftiofur-resistant *E. coli* had only been investigated in one breeder farm, (2) the questionnaire only investigated the types of antibiotics used in the breeder farm, but the administration of antibiotics in which phase was not obtained and (3) the concentration range of different antimicrobials used in the breeder farm was also not included in the questionnaire. Thus, a larger scale of epidemiology with a more detailed questionnaire should be further explored.

## 4. Conclusions

We monitored the prevalence and persistence of ceftiofur-resistant *E. coli* from five stages of layer breeding and found a high prevalence of ceftiofur resistance across the five periods (3.33–100%). In particular, ST101 was the most persistent ST across the four chicken feeding stages and this likely facilitated the high prevalence of these ceftiofur-resistant *E. coli* on the chicken farm. The long-term persistence of ceftiofur-resistant *E. coli* across the layer breeding should be further explored.

## Figures and Tables

**Figure 1 animals-13-00090-f001:**
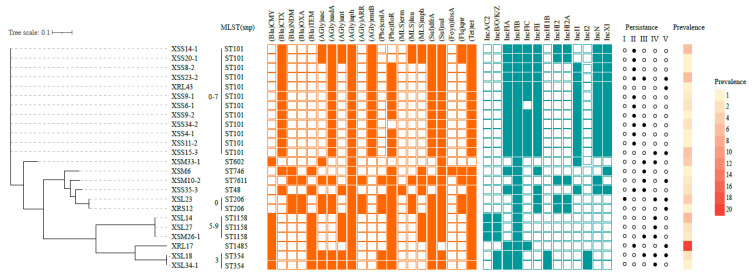
Prevalence, persistence, phylogenetic and genomic characterization analysis of the 24 ceftiofur-resistant *E. coli* isolates we isolated for this study. The orange and green squares represent positivity for ARGs and plasmid Inc types, respectively. Roman numerals represent the persistence of ceftiofur-resistant *E. coli* across five stages of chicken breeding: I, egg; II, GP period; III, EL period; IV, PL period; V, LL period. Filled circles represent ceftiofur-resistant *E. coli* and the heatmap indicates the numbers of ceftiofur-resistant *E. coli* shared between ERIC-PCR patterns.

**Figure 2 animals-13-00090-f002:**
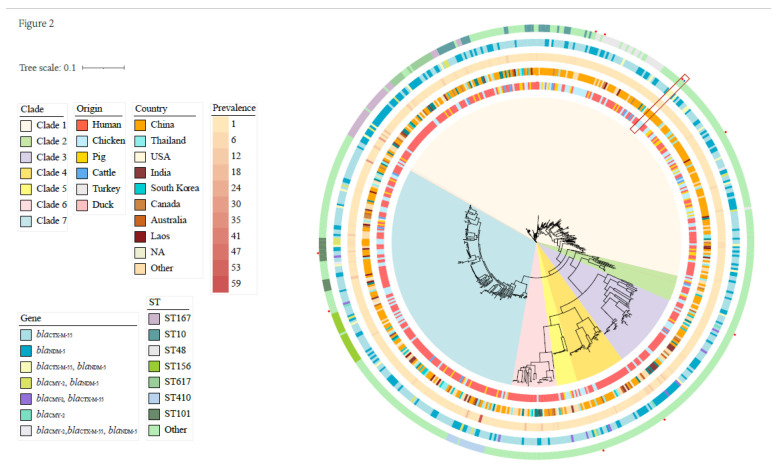
Phylogenetic analysis of the representative ceftiofur-resistant *E. coli* from this study and GenBank. Origins, countries, prevalence, ceftiofur resistance genes and ST types of the isolates are indicated from the inner to outer ring. Red dots represent the isolates from this study and the blue dot represents the isolates from GenBank that share <35 SNPs with an isolate in this study.

## Data Availability

The data will be available with the corresponding author upon request.

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
