# Peer review of "Prevalence and Persistence of Ceftiofur-Resistant Escherichia coli in A Chicken Layer Breeding Program"

_animals, 2022, doi:10.3390/ani13010090_

Round 1

Reviewer 1 Report

Line 2: Please fix E. coli in title.  

Line 58: An experimental design is needed.  A section about animal rearing, description of “stages”, feed, antibiotic treatment, vaccinations, etc.

Lines 59-60: Please expand on the stages of production mentioned here.

Line 62: More details about cloacal sampling procedure are needed.  How long was the swab inserted for?

Line 63: More details about egg sampling procedure are needed. Did you massage the eggshell?  If so, for how long?  Why were the contents of the egg not evaluated?  Sampling the eggshell is more relevant for assessing fecal or environmental contamination.

Line 65: Why was this concentration of ceftiofur used?  

Lines 106-110: The stages of production evaluated are a little confusing.  What do you mean by heat preservation period?  Supplication is also not the correct word to use.

Line 116: Did the flock in the current study ever receive antibiotics?  This will significantly impact the results of this study.  This information must be described in the material and methods.

Line 179-180: This is a strong statement to make without listing any sort of reference. Please remove or alter.  

Line 192-193: Same statement that was listed for lines 179-180

Author Response

Comments and Suggestions for Authors

Line 2: Please fix E. coli in title.

Response: Thank you. We have fixed it (Line 1).

Line 58: An experimental design is needed. A section about animal rearing, description of “stages”, feed, antibiotic treatment, vaccinations, etc.

Response: Thank you. An experimental design (Lines 66-73) and the questionnaire of the antibiotics application (Lines 75-76) in the chicken farm were added in the manuscript.

Lines 59-60: Please expand on the stages of production mentioned here.

Response: Thanks. We have modified the stages of production according to a reference (Lines 48-49 and 68-70)

Line 62: More details about cloacal sampling procedure are needed. How long was the swab inserted for?

Response: Thank you for your reminder. Each cloacal sampling was obtained using the sterile swab that was inserted into the chicken cloaca for at least three seconds. We have modified it in the manuscript accordingly.

Line 63: More details about egg sampling procedure are needed. Did you massage the eggshell? If so, for how long? Why were the contents of the egg not evaluated? Sampling the eggshell is more relevant for assessing fecal or environmental contamination.

Response: Thank you. Each egg was massaged with 8 mL of brain heart infusion (BHI) broth in an aseptic plate for one minute and we have modified it according (Lines 79-80). To explore the prevalence and persistence of ceftiofur-resistant E. Coli in a chicken layer breeding program in current study. A total of 30 eggs were also collected in current study, and we have sampled the Ceftiofur-Resistant E. Coli from the eggshell and one isolate was isolated in current study (Line 126-127).

Line 65: Why was this concentration of ceftiofur used?

Response: Thank you for your reminder. We have checked the record and found the Ceftiofur-Resistant E. Coli was isolated using 8.0 mg L-1 ceftiofur as previously described, and we have modified it in the manuscript (Line 78).

Lines 106-110: The stages of production evaluated are a little confusing. What do you mean by heat preservation period? Supplication is also not the correct word to use.

Response: Thanks. We have modified the stages of production according to a reference (Lines 48-49 and 68-70)

Line 116: Did the flock in the current study ever receive antibiotics? This will significantly impact the results of this study. This information must be described in the material and methods.

Response: Thanks. the questionnaire of the antibiotics application (Lines 75-76) in the chicken farm were added in the manuscript.

Line 179-180: This is a strong statement to make without listing any sort of reference. Please remove or alter.

Response: Thank you for your reminder. We have removed the description.

Line 192-193: Same statement that was listed for lines 179-180

Response: Thank you for your reminder. We have removed the description.

Reviewer 2 Report

Comments:

In this paper, the authors tried to describe the prevalence and persistence of ceftiofur-resistant E. coli in a breeding farm along their production cycle.

To be published needs a major revision and some matters to be clarified:

1.    What do the authors mean by “heat preservation, stopping heat supplication” periods in a breeding farm?

2.    Introduction is very poor with no additional information at all.

3.    Line 34: the authors must specify if ceftiofur is allowed for poultry. In Europe, it is forbidden its use in poultry production.

4.     Lines 37-38:….in zoonotic pathogens and CTX-M-55 is the most
frequently encountered subtype in Escherichia coli poultry infections. The authors must mention in which countries? They must also mention which zoonotic pathogens are they referring

5.    All studies referred to in the introduction were made in broiler farms and the study is in a breeding farm (from lines 41 to 50)

6.    Please explain”… persisted throughout layer breeding excluding the egg period”??  

7.    Single numbers must be written in full, p.e. 5 (five)

8.    Line 18: not “states” but stages

9.    In materials and methods:

§  Sample collection must be separated from the microbiological techniques used for the isolation and characterization of E. coli

§  The laboratory methodology is not properly described. Why did the authors use 16mg/L of ceftiofur if the breakpoint for this drug is 4mg/L (CLSI-VET)? Do the authors understand that many resistant isolates are missed using such a high concentration.

§  How was the microdilution broth method performed. It must be detailed with the antibiotics and concentration intervals.

§  The results were read according to CLSI M100. But ceftiofur is a veterinary drug and there is no breakpoint for ceftiofur in CLSI M-100 (for humans)

§  It is advisable to use use the following bioinformatic tools:

bioinformatics tools of the Center for Genomic Epidemiology: KmerFinder 3.2, to identify the bacterial species, MLST 2.0, to determine the Multilocus Sequence Type, PlasmidFinder 2.1, for the identification of plasmids, ResFinder 4.1, to identify antibiotic resistance genes, SeroTypeFinder 2.0, for serotype identification, CHTyper 1.0 to predict the E. coli FimH type and FumC type, PathogenFinder 1.1 to predict bacteria's pathogenicity towards human hosts, MobileElementFinder v1.0.3 to identify mobile genetic elements and their relation to antimicrobial resistance genes and virulence factors. In addition, ClermonTyping identifies the phylogroup.

10.  Why did the authors selected 1145 E. coli isolates for a phylogenetic tree construction if they have only 24. Were they from poultry, humans?

11.  NDM-5 is a New-Dehli carbapenemase

12. The results of susceptibility testing should be presented in a Table with MIC values obtained.

13. The study was done in one farm, so obviously, the results support some clonal spread

14.  The authors mention in the conclusion: ……one ST746 E. coli isolate of human origin from a commercial chicken farm in Shandong province shared only 26 SNPs with one ST746 isolate in the current study and both carried blaCTX-M-55 and blaNDM-5, suggesting that ST746 E. coli was likely to spread and serve as a potential vehicle to
deliver ARGs between humans and food animals.

§  Was it a human or a poultry isolate?

§  How can the authors infer that the ST746 E. coli was likely to spread and serve as a potential vehicle to
deliver ARGs between humans and food animals.

Round 2

Reviewer 2 Report

Comments:

The correction report was received and analyzed.

Although some corrections were done, the paper still needs some alterations and matters clarified.

1.    The text must be reviewed by an English-speaking person

2.     E. coli in the title is not in full

3.    Several single numbers along the text are not in full

4.    The authors may not generalize as this study was done only on one farm. Nothing is mentioned in the text if the birds received treatment with ceftiofur and other antimicrobials and in which phase.

5.    Regarding bacteriological methodology for the detection of E. coli resistant to 3rd generation cephalosporins, the authors reported in the first version of the manuscript that they used 16mg/l of ceftiofur, and now they refer 8mg/L. In the revision report sent to the authors, I just requested why they used 16 and not 8mg/L. I only needed an explanation and not to change the text. The authors cannot change the methodology.

6.    Nothing is referred about the concentration range of the different antimicrobials used and the breakpoints.

7.    The authors refer a Table 2 (supplementary table) it could not be found. I did not have access to it.

Author Response

We have revised the paper according to the reviewers’ critiques. All of the new changes are highlighted in yellow.  We believe the revised manuscript is improved and we hope it will be acceptable for publication.

  1. The text must be reviewed by an English-speaking person

Response: Thank you. In the last revised version, the language has been totally rephrased by an English-speaking person, and we have revised it in current version (Lines 21 - 23).

  1. coli in the title is not in full

Response: Thank you for your reminder. We have modified accordingly (Line 2)

  1. Several single numbers along the text are not in full

Response: Thank you for your reminder. We have modified it accordingly (Line 17, 39, 41, 175, 182).

  1. The authors may not generalize as this study was done only on one farm. Nothing is mentioned in the text if the birds received treatment with ceftiofur and other antimicrobials and in which phase.

Response: Thank you. We had to acknowledge that there were several limitations in our experimental design, and we have emphasized these limitations in the manuscript to reminder the readers. (Lines 195 - 199) “Our study was limiting in that (1) the epidemiology of the long-term persistence of ceftiofur-resistant E. coli had only been investigated in one breeder farm (2) the questionnaire only investigated the types of antibiotics used in the breeder farm, but the administration of antibiotics in which phase was not obtained (3) the concentration range of different antimicrobials used in the breeder farm was also not included in the questionnaire. Thus, a larger scale of epidemiology with more detailed questionnaire should be further explored.”

  1. Regarding bacteriological methodology for the detection of  coliresistant to 3rd generation cephalosporins, the authors reported in the first version of the manuscript that they used 16mg/l of ceftiofur, and now they refer 8mg/L. In the revision report sent to the authors, I just requested why they used 16 and not 8mg/L. I only needed an explanation and not to change the text. The authors cannot change the methodology.

Response: Thank you for your comment. As for this modification, we have made a mistake. Actually, we isolated the Ceftiofur-Resistant E. coli using 8mg/L, but we followed the wrong experimental notebook carelessly. One proof was the MIC of ceftiofur to nine E. coli isolates, entitled XSM1, XSM26-1, XSL35, XSL22-1, XSL7, XRL22-2, XRL29, XRL38 and XRL41, was 16mg/l. If we used 16mg/l, these isolated were killed in the screening MacConkey plates, and we can not obtain these E. coli isolates

  1. Nothing is referred about the concentration range of the different antimicrobials used and the breakpoints.

Response: Thanks. We have listed the limitation in the manuscript to reminder the readers (Lines 195 - 199).

  1. The authors refer a Table 2 (supplementary table) it could not be found. I did not have access to it.

Response: Thank you for your reminder. We have uploaded supplementary tables in the last revised version. We only investigated the types of antibiotics used in the breeder farm but other information was absent. The table was showed as follow:

Antibacterial administration in the chicken farm

Neomycin Sulphate

Florfenicol

Amoxicillin

Ceftiofur

Tylosin

Oxytetracycline

Doxycycline
